# Resistance of Transgenic Maize Cultivars to Mycotoxin Production—Systematic Review and Meta-Analysis

**DOI:** 10.3390/toxins16080373

**Published:** 2024-08-22

**Authors:** Ana Silvia de Lara Pires Batista Gomes, Saulo Henrique Weber, Fernando Bittencourt Luciano

**Affiliations:** Graduate Program in Animal Science, School of Medicine and Life Sciences, Pontifícia Universidade Católica do Paraná, 1155 Imaculada Conceição Street, Prado Velho, Curitiba 80215-901, Brazil; anaslpbg@gmail.com (A.S.d.L.P.B.G.); saulo.weber@pucpr.br (S.H.W.)

**Keywords:** crop production, food safety, aflatoxin, deoxynivalenol, fumonisin, zearalenone

## Abstract

Approximately 25% of cereal grains present with contamination caused by fungi and the presence of mycotoxins that may cause severe adverse effects when consumed. Maize has been genetically engineered to present different traits, such as fungal or insect resistance and herbicide tolerance. This systematic review compared the observable quantities, via meta-analysis, of four mycotoxins (aflatoxins—AFL, fumonisins—FUM, deoxynivalenol—DON, zearalenone—ZEA) between genetically modified (GM) and conventional maize kernels. This study was conducted following the PRISMA guidelines, with searches performed using PubMed, Web of Science, Scopus, Google Scholar, and CAPES journals databases. Analyses were conducted using RevMan v.5.4 software. Transgenic maize showed a 58% reduction in total mycotoxins (*p* < 0.001) compared to conventional maize. FUM were the most impacted, with a 59% reduction (*p* < 0.001) in GM maize. AFL and ZEA levels were also lower in GM maize by 49% (*p* = 0.02) and 51% (*p* < 0.001), respectively. On the other hand, DON levels increased by 6% (*p* < 0.001) in GM maize compared to conventional maize. However, results for ZEA and DON were inconclusive due to the limited research and sample sizes. We conclude that transgenic maize reduces total mycotoxins by over 50%, primarily fumonisin and aflatoxin. Most studies presented maize varieties that were resistant to insects or herbicides, not fungal pathogens, showing a positive collateral effect of these genetic alterations. Therefore, transgenic maize appears to be a safer product for animal and human consumption from a toxicological point of view. Further studies with larger sample sizes are needed to confirm our findings for ZEA and DON in transgenic maize.

## 1. Introduction

Fungal contamination affects approximately 25% of grains consumed by humans and animals [1]. *Alternaria*, *Aspergillus*, *Fusarium*, and *Penicillium* species are commonly found in these crops and produce secondary metabolites presenting low molecular weight and toxic effects, known as mycotoxins [1]. Over the past few decades, more than 300 mycotoxins have been identified that pose risks to human and animal health, such as neurological disorders, pulmonary edema, alteration of endocrine functions, and cancer [1,2]. The primary mycotoxins found in contaminated animal feed include aflatoxins (AFL), deoxynivalenol (DON), fumonisins (FUM), ochratoxins, zearalenone (ZEA), and patulin [3]. The elimination of these substances from food is difficult because of their chemical characteristics, which confer resistance to the processing stages [4]. In the United States, annual losses in maize production due to AFL contamination alone are estimated to be USD 52.1 million to USD 1.68 billion [5]. For these reasons, several countries have legislated and enforced maximum levels of mycotoxins in certain grains, determining the limits allowed in products destined for animal consumption (Table 1).

In general, mycotoxins ingested by humans and animals, particularly dairy cattle, poultry, horses, pigs, and fish, can cause acute toxicity (hemorrhages, abortions, gastroenteritis), neurological disorders, alterations in nutritional content, alterations in endocrine and neuroendocrine function, suppression of the immune system, cancer, decreased conception rate, pulmonary edema, and leukoencephalomalacia, with consequent negative economic impacts in public health and animal food production [11,12].

The main mycotoxins found in maize during plant growth are FUM, DON, and ZEA, which are mainly produced by pathogenic species of *Fusarium* [13]. These mycotoxins can occur concomitantly and act synergistically, increasing the chances of toxicological problems [14]. The most common mycotoxin produced during grain storage is AFL, which is derived from the secondary metabolism of *Aspergillus* and *Penicillium* species. AFL is a potent carcinogen for humans and animals described as a class 1 carcinogenic compound by the International Agency for Research on Cancer [2,13]. AFL can also be found in the field, to a lesser extent, in plants damaged by drought or pests [14,15].

One possible method to mitigate fungal toxins in grains is the development of transgenic plants that resist the growth of phytopathogenic fungi and can consequently inhibit the growth of toxigenic fungi and their production of secondary metabolites. The term transgenic refers to a type of genetically modified organism that has received exogenous genetic material and can express a desired trait [16]. Maize, soybeans, oats, rice, sorghum, and barley stand out among the important grains in animal feed [17]. Maize is the second highest-produced grain in the world, with approximately 50% of its production destined for animal consumption [18]. Transgenic maize cultivars have been developed for different purposes, including resistance to insects and herbicides [19], and some cultivars are resistant to *Fusarium* [20,21]. Insects and weeds may damage maize plants, promoting the growth of opportunistic fungal species. Genetically modified cultivars maintain productivity and nutrient content comparable to non-transgenic varieties [22] and prevent fungal growth, thus reducing mycotoxin production [23].

Studies have compared transgenic and non-transgenic maize regarding mycotoxin production [24,25], yet a comprehensive overview identifying effective transgenic traits and the specific mycotoxins mitigated is lacking. Therefore, the present systematic review aims to analyze the effects of transgenic maize in mitigating FUM, AFL, DON, and ZEA through a meta-analysis. The research question used for this study was, “Do genetically modified maize cultivars have the ability to mitigate mycotoxins compared to conventional cultivars?”.

## 2. Results

This section was divided into general and specific analyses of mycotoxin quantities, including both the total analysis of all mycotoxins combined and the analysis of each toxin individually. For mycotoxin concentration, the groups were categorized based on the average mycotoxin levels in non-transgenic maize.

### 2.1. General Findings

Based on the selection criteria and subsequent application of the exclusion criteria by title, 325 articles were selected. A total of 178 articles were further selected based on their abstracts and methodologies. Ultimately, 45 articles [21,26,27,28,29,30,31,32,33,34,35,36,37,38,39,40,41,42,43,44,45,46,47,48,49,50,51,52,53,54,55,56,57,58,59,60,61,62,63,64,65,66,67,68,69,70] were included in the systematic review (Appendix A).

The US had the highest number of studies (eighteen) [26,27,28,39,40,42,44,45,49,50,53,54,56,60,64,66,67,68], followed by Brazil, Italy, Germany, and Spain, with five [21,32,38,43,47], four [28,33,35,36], three [37,57,65], and three [29,30,51] studies, respectively. Additionally, two studies were conducted each in South Africa [58,61], Argentina [41,59], and France [30,46]. Isolated studies were carried out in Canada [62], Hungary [31], Japan [52], Kenya [55], Mexico [34], the Philippines [41], China [69], and the Republic of Korea [63]. Data regarding planting season, cultivation method, and the number of hybrids per study are presented in Table 2.

The number of samples analyzed per article ranged from 1 to 480, with an average of 35 samples, consisting of grain pools, ground grains, or maize residues (leaves, stems, and ears with and without grains).

The primary purpose of genetic modification was to confer resistance to insects, which was found in 23 cases (51.1%), followed by herbicide resistance in 11 cases (24.4%), transformations for *Aspergillus flavus* resistance in 5 cases (11.1%), and antibiotic resistance in 2 cases (4.4%). From the total number of studies evaluated, 31.1% (14) did not provide data on the technique or function of the genetic modification (Table 3).

The mycotoxins analyzed in the present study were AFL, FUM, DON, and ZEA. Ochratoxin and nivalenol were also evaluated in a few studies included in this review but were not subject to our analyses. Based on the collected data, a high number of studies were found for FUM (71.1%), followed by AFL (44.4%), DON (29%), and ZEA (24.4%) (Figure 1). Three articles analyzed the quantity of all four mycotoxins, observing both their isolated and concomitant presence. Three other articles evaluated both FUM and AFL, four evaluated DON, FUM, and ZEA, two evaluated AFL and DON, and only one evaluated DON and ZEA. No article evaluated AFL together with DON or ZEA, and likewise, no article evaluated FUM together with DON or ZEA.

The main methods used for the quantification of mycotoxins were liquid chromatography with fluorescence detection (FLD), enzyme-linked immunosorbent assay (ELISA), and liquid chromatography coupled with mass spectrometry (MS/MS), which were used in 23, 15, and 16 studies, respectively. In addition, rapid one-step assay (ROSA), VICAM AflaTest, Fungi-Plex multiplex mycotoxin assay kit, FluoroQuant AflaTest, and Ridascreen Fast Test were used. Six articles did not describe the quantification methodology. In some cases, more than one test was used in the same study to evaluate one or more mycotoxins (Figure 2 and Table 3).

Subsequently, the mean concentration values of mycotoxins in maize, standard deviations, sample sizes, and *p*-values of the analyses were obtained (Table 4). Some studies have assessed more than one mycotoxin, and multiple experiments have been conducted in the same study. Therefore, thirteen studies contained a total of 33 experiments. 

Thirteen articles were included in the meta-analysis, containing 33 experiments; all evaluated transgenic hybrids having resistance to insects, with six out of thirteen (46%) being resistant only to insects, six out of thirteen (46%) also presenting resistance to herbicides, and one (8%) expressing resistance to herbicides and antibiotics (Table 5). Therefore, resistance to insects might also contribute to the decrease in mycotoxin quantities in transgenic grains.

Of the thirty-three experiments, eight showed relatively high amounts of mycotoxins in transgenic plants, with three out of eight (38%) [39,46] showing resistance to insects only and the other five (62%) [28,44,57,66] showing resistance to insects and herbicides. 

The results obtained in the meta-analysis were shown in the forest and funnel plots presented in the Appendix A, in which it was possible to observe the symmetry of the results and the low dispersion. The funnel plot is used for assessing publication bias, thus observing the tendency for statistically significant studies to be more likely to be published [70]. Additionally, the dispersion of points indicates the precision of the studies, in which studies with higher precision exhibit lower dispersion, while studies with lower precision exhibit higher dispersion [70].

### 2.2. Mycotoxin Levels and Groups Division

A significant difference (*p* < 0.001) was observed in the total amounts of FUM, AFL, DON, and ZEA in 1846 samples between non-transgenic (*n* = 954) and transgenic (*n* = 892) maize (Appendix A). To understand the discrepancies and similarities in detail between the studies developed in 33 experiments [21,26,27,28,38,39,44,46,51,57,61,66,69], the following three groups were considered: average concentrations < 10 mg/kg, between 10 and 100 mg/kg, and >100 mg/kg.

The first group consisted of 15 experiments (46%) divided into seven articles [26,27,28,39,44,46,66] with mean concentrations of mycotoxins ranging between 0.02 and 6 mg/kg, containing at least 1 experiment for each evaluated mycotoxin. Of the 15 experiments, 14 (93%) [26,27,39,44,46,67] used grain samples taken from maize harvested between the summer and autumn seasons, whereas 1 experiment [28] used samples of subsequently inoculated maize residue. Five out of fifteen experiments (33%) evaluated FUM [39,44,66], six (40%) evaluated AFL [26,27,28,66], two (14%) [46] evaluated DON, and two (14%) [46] evaluated ZEA; here, 10 experiments [26,27,28,46,66] used high-performance liquid chromatography with mass spectrometry (HPLC–MS/MS) for quantification, and five (33%) [39,44] used HPLC with FLD (HPLC–FLD). Regarding the mean amounts of mycotoxins, six experiments (40%) [27,39,44,46] presented a higher amount of mycotoxins in transgenic plants than in non-transgenic plants, one (7%) [66] presented similar values for both groups, and eight (56%) [26,28,39,44,46,66] presented lower means in transgenic plants than in non-transgenic plants; this group was positively conclusive.

The second group consisted of eight experiments (24%) divided into five articles [26,39,51,57,69] with means ranging between 3 and 53.44 mg/kg, in which six experiments (75%) evaluated FUM [26,39,51,69] and two (25%) evaluated DON [57], with all samples taken from maize harvested between spring and autumn. Immunoassay detection methods were used in three (38%) experiments (ELISA in two [57] and ROSA in one [51]), three (38%) used HPLC–MS/MS [69], and HPLC–FLD was used in the other two experiments (24%) [26,39]. In seven out of eight experiments [26,39,51,57,69], the mean amounts of mycotoxin were lower in transgenic than in non-transgenic maize. The opposite results were observed in the other one [57]. Despite the opposing results observed in these studies, considering the sample size and methodological quality described in the articles, it can be concluded that this result is conclusive; this means that the concentration of mycotoxins in transgenic maize was lower than in non-transgenic maize.

The third group consisted of ten experiments (30%) divided into six articles [21,26,38,46,61,69] with means ranging between 5 and 820 mg/kg. All experiments involved FUM analysis, with seven experiments (70%) [21,26,38,46,61] using HPLC–FLD for quantification and the other three (30%) using HPLC–MS/MS [69]. All average concentrations of mycotoxins were significantly lower in the transgenic group than in the non-transgenic group, thereby rendering this group positively conclusive. 

Among the 33 experiments analyzed, 3 (9%) [26] presented discrepant values for mycotoxin concentration, prompting a new analysis without those experiments. Therefore, the total number of experiments decreased from 33 to 30 [26,27,28,39,44,46,61,66,69], and that of the first group decreased from 15 to 12 articles. In this analysis, the first group had six experiments (50%) [27,39,44,46] with higher amounts of mycotoxins in transgenic groups than in non-transgenic groups; one (8%) [66] showed equal concentrations for both groups and five (42%) [28,39,44,46,66] showed lower mean amounts in transgenic groups than in non-transgenic groups, thereby rendering the results of this group inconclusive. No changes were observed in the other two groups, and the results showed a significant difference (*p* < 0.001) in mycotoxin amounts with a positive and conclusive outcome (Appendix A).

### 2.3. FUM Comparison in Conventional and Transgenic Maize

A significant difference (*p* < 0.001) in the amount of FUM was observed between transgenic (*n* = 726) and non-transgenic (*n* = 761) maize in 21 experiments [26,38,39,44,46,51,61,66,69], accounting for 64% of 33 experiments (Appendix A). To understand the discrepancies and similarities among the studies in detail, the following three groups were considered: mean values < 10 mg/kg, between 10 and 200 mg/kg, and >200 mg/kg. 

The first group was composed of five experiments (24%) [39,44,66] with mean amounts of FUM ranging from 0.5 to 4.8 mg/kg extracted from maize samples directly harvested from the field and quantified using HPLC–FLD. Three [44,66] of the five experiments (60%) analyzed AFL along with FUM. Regarding the amount of FUM, two (40%) experiments [39,44] showed higher mean values in transgenic groups than in non-transgenic groups, two (40%) [39,44] showed lower mean values in transgenic groups than in non-transgenic groups, and one (20%) [66] presented the same mean value for both groups and was inconclusive.

The second group, with 10 of the 21 experiments (48%) [26,39,46,51,61,69], had mean values of mycotoxin ranging between 12.48 and 141.12 mg/kg. Seven experiments (70%) [26,46,69] used HPLC–MS/MS as the quantification method, two (20%) [39,61] used HPLC–FLD, and one (10%) used ROSA [51]. Seven experiments (70%) [39,51,61,69] evaluated FUM only, whereas three (30%) [26,46] evaluated other mycotoxins. All experiments showed higher mean values in non-transgenic groups than in transgenic groups; therefore, this group was positively conclusive.

The third group consisted of 6 of the 21 experiments (29%) [21,26,38,61] with average FUM levels ranging between 255.2 and 820 mg/kg in samples of ears harvested in the field. Three of the six experiments (50%) [26] evaluated AFL along with FUM, and the other three [12,38,61] evaluated FUM only. All experiments used HPLC–MS/MS to detect mycotoxins. All experiments showed lower averages in transgenic groups than in non-transgenic groups; therefore, this group was positively conclusive.

Among the 21 analyzed experiments, 3 (14%) [26] showed highly different mycotoxin levels, resulting in a new analysis without these experiments. Therefore, the total number of experiments decreased from 21 to 18 [26,38,39,44,46,51,61,66,69], and that of group three decreased from 6 to 3 experiments [38,62]. Group three showed lower averages in transgenic groups than in non-transgenic groups, rendering the results of this group positively conclusive. No change for the other two groups was observed, and the results showed a significant difference (*p* < 0.001) in mycotoxin levels and had a positively conclusive outcome (Appendix A). Therefore, based on the evaluated samples, the amount of FUM in GM maize was lower than that in non-GM maize.

### 2.4. AFL Levels in Conventional and Transgenic Maize

In the meta-analysis, a difference (*p* = 0.02) was observed in the amount of AFL between non-transgenic (*n* = 61) and transgenic (*n* = 34) maize in 6 of the 28 experiments [26,27,28,66], comprising 21% of the total experiments (Appendix A). To better understand the similarities and discrepancies between studies, the following two groups were formed: those with average doses <1 mg/kg and those with average doses ≥1 mg/kg.

The first group consisted of three out of six experiments (50%) [26,28,66] with average amounts of mycotoxin ranging between 0.04 and 0.4 mg/kg quantified through fluorescence detection. Two out of three experiments (67%) were conducted with maize samples directly harvested from the field [26,66], and one (33%) used samples of maize residues (ears with and without grains, leaves, and stems) collected from the field and inoculated with *A. flavus* isolates. Two experiments (67%) evaluated FUM and AFL, and all samples were collected during late summer/early autumn. Two experiments (67%) showed higher average amounts in transgenic maize than in non-transgenic maize, thereby presenting a negatively conclusive result for this group. 

The second group contained three experiments (50%) [26,27] with average amounts ranging between 1 and 4 mg/kg. All experiments were conducted with maize samples collected directly from the field in the same manner as in the previous group, during late summer/early autumn. The fluorescence-based quantification and analysis of one or more mycotoxins, along with AFL, were unanimously used. All experiments in this group showed lower mycotoxin amounts in transgenic maize than in non-transgenic maize, resulting in a positively conclusive result for this group.

In Group 1, the amount of AFL was higher in transgenic maize than in non-transgenic maize, whereas the opposite trend was observed in Group 2. Although a significant effect was observed in the meta-analysis, the results were inconclusive.

### 2.5. DON Levels in Maize

In the meta-analysis, a significant difference (*p* < 0.001) was observed in the levels of DON between transgenic (*n* = 90) and non-transgenic (*n* = 90) maize, with 4 out of 28 (14%) [46,57] experiments showing average DON levels between 2 and 43 mg/kg (Appendix A).

Four experiments were conducted using field-harvested ear samples. Two experiments (50%) [57] evaluated FUM and ZEA in addition to DON and quantified them using HPLC–MS/MS, whereas the other two (50%) [46] quantified DON only using immunochromatographic ELISA. Of the four experiments, two (50%) [46] showed higher average levels of mycotoxins in transgenic maize than in non-transgenic maize; one (25%) [57] showed equal levels in both groups, and one (25%) [57] showed lower levels of mycotoxins in transgenic maize than in non-transgenic maize.

Therefore, despite the significant effect observed in the meta-analysis, the results were inconclusive.

### 2.6. ZEA in Non-Trangenic and Trangenic Maize

In the meta-analysis, a highly significant difference (*p* < 0.001) was observed in the amount of ZEA between non-transgenic (*n* = 42) and transgenic (*n* = 42) maize in 2 out of 28 experiments (7%) [46], with mean values ranging between 0.02 and 0.4 mg/kg (Appendix A).

Both experiments were conducted in France, with maize ears harvested in the field during summer. Mycotoxins were quantified using mass spectrometry (MS/MS). Other mycotoxins were also evaluated using the same samples, and the mean levels of mycotoxins were lower in transgenic maize than in non-transgenic maize.

Therefore, despite a significant effect observed in the meta-analysis, the results were inconclusive, owing to the number of experiments.

Based on these results, a significant difference (*p* < 0.05) in the levels of mycotoxins was observed during the evaluation of total and individual mycotoxins. These findings indicate that, except for DON, higher amounts of mycotoxins were present in conventional maize than in transgenic maize.

## 3. Discussion

In the analysis of overall mycotoxin content, transgenic maize exhibited a significant reduction by 58% (*p* < 0.001) with respect to conventional maize. Similar outcomes have been observed for the levels of FUM, with a 59% reduction (*p* < 0.001) in GM maize. Additionally, the levels of AFL and ZEA were lower in GM maize than in conventional maize, with reductions of 49% (*p* = 0.02) and 51% (*p* < 0.001), respectively. Conversely, DON levels showed a significant increase by 6% (*p* < 0.001) in GM maize with respect to those in conventional maize. Despite the high occurrence of AFL during storage, evidence supports its production in the field, which may cooccur with the production of other mycotoxins, such as FUM, DON, and ZEA in the field, allowing for an equivalent comparison of the effects of transgenic maize for all analyzed mycotoxins [14]. These results are supported by the forest and funnel plots obtained from the meta-analysis. Observing the result of the forest plot is important because, in the absence of bias, the distribution of points on the graph is symmetric around the central line, while in the presence of bias, this dispersion of points is wider and asymmetric [70]. Additionally, the asymmetry of the funnel plot can be influenced not only by bias but also by the precision of the studies used [70].

The higher relative difference found in the quantities of mycotoxins was observed in the group with higher concentrations of fumonisin. This demonstrates that specifically in the group that causes the highest economic losses, the best results were observed. The results presented in groups with higher means showed a relatively greater reduction compared to groups with lower means, which may be associated with the fact that insect damage levels are higher. However, there are no studies that have evaluated the correlation between different insect infestation levels and the production of mycotoxins. In order to establish this correlation, a controlled study would be necessary, with different insect infestation levels, to assess the extent of damage to maize plants and subsequently measure the amount of mycotoxins produced.

Damage by insects, drought, and other factors that can cause tissue damage in plants are among the factors that serve as gateways for fungal contamination and subsequent mycotoxin production [14,61,71]. The use of transgenic maize allows for the reduction of target insects in the field without affecting other insect populations [72]. This is owing to the use of genes, including cry1A and cry1Ab, which are toxic to specific insect species [73]. These genes belong to *Bacillus thuringiensis*, the most common insertion in transgenic maize, and are toxic to insects, particularly those in the Coleoptera, Diptera, and Lepidoptera families that are harmful to maize and other crops [73]. When the proteins produced by cry1A and cry1Ab are ingested by insects, they are dissolved by proteases and the alkalinity of their intestines and form toxic crystals that result in inflammation, starvation, and the eventual death of the insects [74]. Insects from the three main families mentioned above cause damage to plants by chewing, which primarily affects roots, leaves, and stems [75]. These insects act as vectors of infections, and the damage they cause paves the way for contamination by pathogens, mainly fungi and bacteria [75].

In this way, once insect-resistant transgenic maize reduces the number of insects, it may subsequently reduce damage to maize, allowing for reduced fungal contamination and lower mycotoxin production [71], which is in line with the findings of the present study, providing evidence for the reduction of mycotoxins in insect-resistant transgenic maize kernels. When insects attack the outer protective layers of maize kernels, nutritious elements are exposed at high humidity and temperature, which is ideal for fungal growth.

Regarding herbicide-resistant transgenic maize, the effect of herbicides on fungal contamination has not been observed; however, the lack of use of herbicides impacts the production of mycotoxins because certain weed species can facilitate fungal growth, which can subsequently contaminate maize [76]. In maize, the use of herbicides (acetochlor, sulcotrione, metribuzin, and pendimethalin) reduces weed competition and decreases competition for nutrients, water, and light, allowing for an increase in the percentage of carbohydrates in grains without leaving residues in plants [77]. Glyphosate reduces weed competition and entails plant growth; however, its use is toxic to plants, affecting the accumulation of stalk dry mass and plant height, causing high accumulation and growth at low doses, but the reverse effect is observed at high doses [78]. Notably, the level of toxicity in plants exposed to glyphosate is dose-dependent. At a dose of 72 g·ha-1, low levels of toxicity (1.379) are noticed. When glyphosate is combined with phosphite (3 L·ha^−1^), the level of toxicity in plants increase up to 16.3% at a dose of 72 g·ha-1 glyphosate, leading to a significant reduction in plant height by 54% [78]. Given these circumstances, the use of herbicide-resistant transgenic maize prevents weed competition, thereby reducing the proliferation of fungi that use weeds as vectors and consequently preventing contamination by fungi and subsequent production of mycotoxins.

In the articles included in this systematic review, there were different geographical regions where the experiments were conducted. A greater number of studies were observed in the Northern Hemisphere compared to the Southern Hemisphere, where differences in temperature, precipitation, and other factors that influence fungal growth and mycotoxin production are significant. The relation of environmental parameters was analyzed previously and confirmed in a systematic review that studied the distribution of mycotoxins worldwide [79]. Temperature plays an important role in fungal contamination and mycotoxin production [80]. The optimal temperature for AFL, FUM, DON, and ZEA ranges between 25 and 30 °C; however, mycotoxin production can occur to a lesser extent at temperatures between 4 and 37 °C [79]. Some fungal species, such as *Fusarium* species seem to dominate in more temperate climates, whereas *Aspergillus* is more commonly found in warmer weather. In general, locations with higher temperatures and humidity will promote higher mycotoxin production.

Among the 45 articles included in the review, 3 [31,48,61] presented an uncertain risk of bias, according to Cochrane bias guidelines, while the others presented a low risk. The observed risk was due to sample allocation, blinding, and incomplete or selective outcomes. However, these articles contained relevant information for the review and contributed positively to the development of the research and were therefore retained in the present review once they presented results consistent with those of the other articles.

## 4. Conclusions

Based on the results obtained from the systematic review and meta-analysis, it can be concluded that (i) for the total amount of mycotoxins, fumonisin, and aflatoxin, genetically modified maize produces a lower amount of mycotoxin compared to non-genetically modified maize (positively conclusive); (ii) for zearalenone and deoxynivalenol, the results are inconclusive.

We conclude that the use of transgenic maize reduces over 50% of the amount of mycotoxins, in which fumonisin and aflatoxin production in transgenic maize were greatly inhibited, generating safer grains from a toxicological standpoint. Further studies involving zearalenone and deoxynivalenol are needed to confirm their reduction in transgenic maize, offering a larger sample size for a conclusive observation.

## 5. Materials and Methods

This systematic review was conducted using Preferred Reporting Items for Systematic Reviews and Meta-Analyses (PRISMA) (Appendix A) [81], following the PICO acronym standard, as follows: population, intervention, control, and outcome. Briefly, the search terms are defined according to the research question that the review aims to answer. Then, an initial search is conducted to assess the feasibility of conducting the review according to the PRISMA checklist. After the final determination of the terms and feasibility of the research, an exhaustive search for records is performed. The studies obtained in this search are screened by file type, language, quality of the published journal, year of publication, title, and abstract according to the inclusion and exclusion criteria. The resulting articles are then included in the review.

The concentrations of FUM, AFL, DON, and ZEA in genetically modified and conventional maize grains were analyzed, with maize as the population and mycotoxin quantity as the outcome. Once this systematic review aimed to evaluate the ability of transgenic methods in maize to mitigate economically relevant mycotoxins, a comprehensive search was performed using the following search protocol:

Population: maize OR corn OR Zea mays

AND

Intervention: transgenic OR transgenic plant OR OGM

AND

Control: conventional maize OR conventional corn

AND

Outcome: mycotoxin OR fungal toxin

The search was conducted using PubMed, Web of Science, Scopus, Google Scholar, and the CAPES (Brazilian Higher Education Personnel Improvement Coordination) journal portals. The initial search yielded 2597 articles. English was the language used for the search. Only articles published in scientific journals using Qualis A1 to B2 (according to CAPES classification) or SCIMago Q1 and Q2 were included. The searches were conducted from February to April 2024.

From the articles selected using the search protocol, the title, year, journal, journal quality, and database were recorded.

The exclusion criteria were as follows: (i) duplicate articles (same article in different databases); (ii) title (articles without maize as a population, evaluating pathogens and not mycotoxins, or not containing transgenics); (iii) abstract (articles that did not have mycotoxins as an outcome, did not contain transgenic maize, or were literature or systematic reviews); and (iv) methodology (absence of average mycotoxin levels, described quantification techniques, and description of samples used).

After applying the exclusion criteria, 45 articles [21,26,27,28,29,30,31,32,33,34,35,36,37,38,39,40,41,42,43,44,45,46,47,48,49,50,51,52,53,54,55,56,57,58,59,60,61,62,63,64,65,66,67,68,69] were selected (Figure 3), to which an extraction form was applied, containing the analyzed mycotoxin, amount of mycotoxin, unit of quantification, sample value, methodology for quantification, region where experiments were conducted, crop mode and season of planting and harvesting, cultivated hybrids, type of transgenics, and proteins and genes used, when described.

Articles with compatible methods for quantifying mycotoxins were considered suitable for extraction, as well as those that presented the evaluated maize hybrids and described the geographic region and execution of the experiments, listed the evaluated mycotoxins and their quantities, described the proteins used in the process and genes that conferred resistance, and provided the sample size (n) of cultivars used.

The risk of analysis bias was performed according to the Cochrane method [83]. Risk can be low, high, or uncertain and includes the following five points: sample allocation, blinding, incomplete or selective outcomes, sample size, and journal quality. A study is considered low-risk for sample allocation when a random selection or random number generator is used to select the samples, and it is considered high-risk when the selection is performed. Regarding blinding, it is considered low-risk when the test is double-blind, and it is considered high-risk when it occurs without blinding, which may be owing to evaluation bias or the placebo effect. Outcomes present a low risk when no loss of samples is observed or the loss is justified and a high risk in the case of many unexplained losses. A low risk for sample size is associated with numerous samples, whereas a high risk is associated with few samples. Finally, the journal quality is evaluated, with a low risk for Q1 and Q2 journals and a high risk for Q4 or non-indexed journals.

### Statistical Analysis

Data were compiled and organized for meta-analysis using Review Manager v.5.4 [71]. The amount of mycotoxin (mg/kg) was compared between conventional and transgenic maize using the standard mean difference with 95% confidence intervals and a fixed-effect model. Heterogeneity was evaluated using I² and Cochran’s Q tests, and the overall effect between treatments was analyzed using the Z-test. A *p*-value < 0.05 was considered statistically significant. Analyses were conducted for both the total amount of mycotoxins (evaluating the four mycotoxins together) and each mycotoxin individually.

The results were classified as conclusive or inconclusive based on inverse variance analysis and sample size. Conclusive results were described as having significantly different values, as well as high sample size intensity. Positively conclusive results were those in which transgenic grains had relatively low levels of mycotoxins compared with non-transgenic grains and were negatively conclusive when transgenic grains had relatively high levels of mycotoxin compared with non-transgenic grains. Inconclusive results were those that did not show significant agreement (i.e., opposite results regarding the mean values of transgenic and non-transgenic maize) or had low sample size intensity.

Figures and images for visual summarization were prepared using the Lucidchart^©^ software [84].

The textual content of this article has been grammatically reviewed by Editage.

This review was not registered and had no prepared protocol.

## Figures and Tables

**Figure 1 toxins-16-00373-f001:**
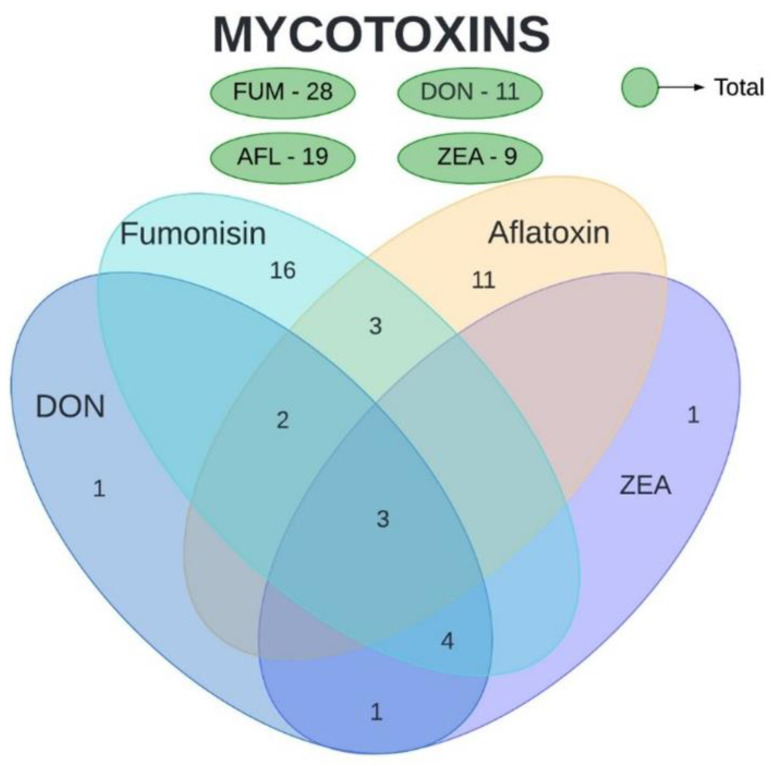
Number of articles evaluating mycotoxin concentration in maize. Total number of articles per mycotoxin is presented in green. Intersections represent the number of studies that shows the concomitant evaluation of mycotoxins. AFL—aflatoxins. DON—deoxynivalenol. FUM—fumonisins. ZEA—zearalenone.

**Figure 2 toxins-16-00373-f002:**
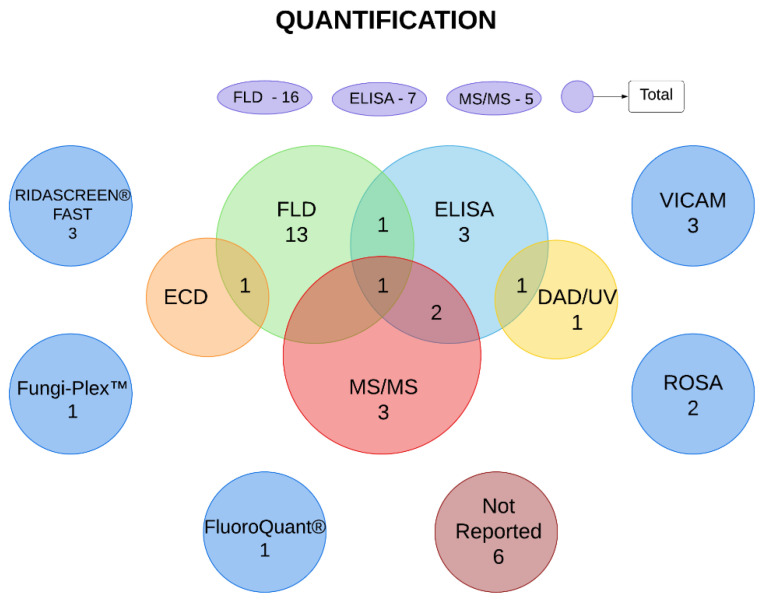
Methods of mycotoxin quantification used in the selected articles. Numbers presented in the oval/circular elements represent the number of studies that used determined method. Intersections represent the number of studies that used two or more methods to analyze mycotoxins. The figure was created using Lucidchart^©^ software (online version www.lucidchart.com (accessed on 14 September 2022)). DAD/UV—diode array detector/ultraviolet. ECD—electron-capture dissociation. ELISA—enzyme-linked immunosorbent assay. FLD—fluorescence detection; MS/MS—tandem mass spectrometry. ROSA—rapid one-step assay.

**Figure 3 toxins-16-00373-f003:**
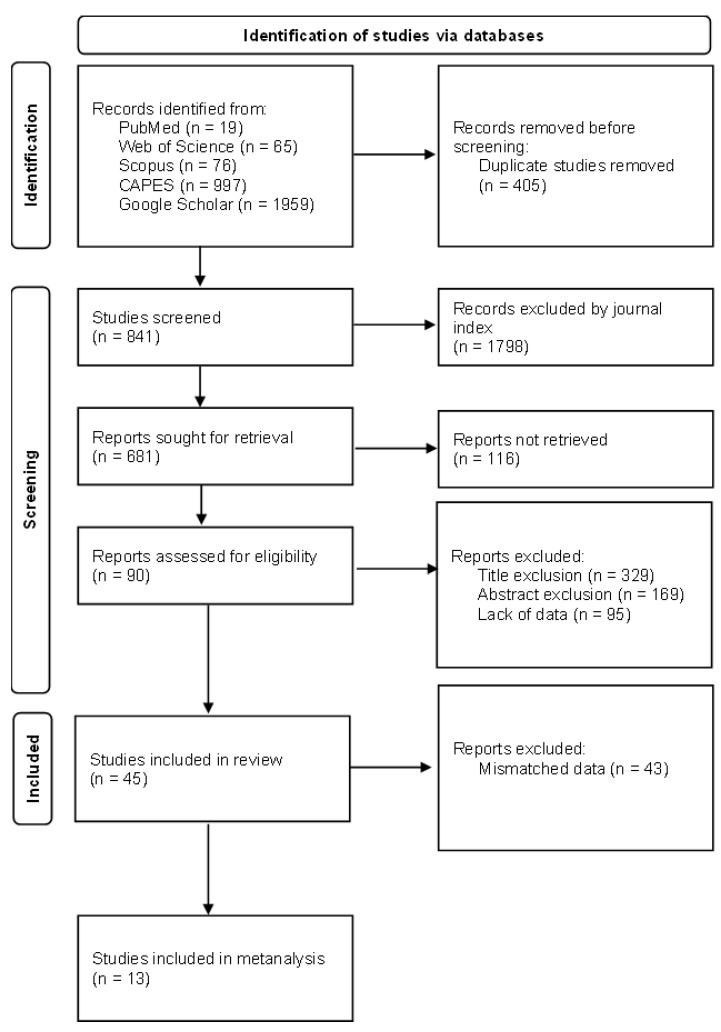
Flow diagram of study selection based on PRISMA [82].

**Table 1 toxins-16-00373-t001:** Maximum concentration of mycotoxins permitted in maize and maize-derived products used for animal production in major crop-producing countries. Data presented are specific for maize intended for animal feed.

Country/Region	Mycotoxin	Maximum Level Allowed (μg/kg)
Argentina [6]	AFL	20
Brazil [7]	AFL	50
China [8]	AFL	50 ^1^
	FUM	60,000
	ZEA	500
Europe Union [9]	AFL	20
	FUM	60,000
	DON	12,000 ^1^
	ZEA	3000 ^1^
United States [10]	AFL	20–300 ^2^
	FUM	5000–30,000 ^2^
	DON	5000–10,000 ^2^

^1^ Maize products only; ^2^ maximum tolerated values may vary depending on the intended use of maize in animal production. AFL—aflatoxin. DON—deoxynivalenol. FUM—fumonisins. ZEA—zearalenone.

**Table 2 toxins-16-00373-t002:** This table presents data on the number of articles for each season and cultivation method, as well as the number of hybrids used in each study. Cultivation took place in the summer, spring, and autumn, and the cultivation methods were as follows: fields, greenhouses, growth chambers, glasshouses, and material collection after harvest. Regarding the hybrids, a total of 277 conventional and transgenic hybrids were analyzed across 1498 samples, with each study analyzing between 1 and 43 hybrids. The hybrids studied included conventional maize, crosses, genetically modified organisms, and transgenic plants.

Planting Season	
**Season**	**Number of Articles**
Summer	28 [21,26,28,29,30,32,33,37,38,40,41,43,44,45,46,47,48,50,53,56,57,58,61,62,65,66,67,69]
Spring	10 [21,27,35,36,39,51,59,68,70]
Autumn	10 [26,27,28,33,34,35,36,39,40,51]
Artificial temperature control	3 [34,54,63]
Not reported	8 [31,42,48,49,52,55,60,64]
**Cultivation Method**	
**Method**	**Number of articles**
Field	34 [21,26,27,29,30,32,33,35,36,37,38,39,40,41,43,44,45,46,47,48,50,51,53,56,57,58,59,61,62,65,66,67,68,69]
Greenhouse	4 [49,52,63,64]
Growth chamber	2 [54,60]
Glasshouse	2 [42,55]
Material collected after harvest	2 [28,34]
Not reported	1 [31]
**Maize hybrids**	
**Number of hybrids**	**Number of articles**
1	7 [29,45,51,52,55,60,67]
2	8 [27,30,31,35,36,38,46,65]
3	6 [28,33,39,43,58,66]
4–10	10 [21,32,44,48,49,56,57,62,63,64]
>10	7 [26,34,37,41,42,47,50]
Not reported	4 [40,48,53,61]

**Table 3 toxins-16-00373-t003:** Number of articles used in the systematic review per transgenic trait.

Transgenic Objective	Number of Articles
Insect resistance/tolerance	23 [21,26,27,28,30,31,34,38,39,41,44,46,48,50,51,53,56,57,58,61,65,66,67,69]
Herbicide resistance/tolerance	11 [28,38,44,48,52,57,61,64,66,67,69]
*Aspergillus flavus* resistance	5 [42,49,54,55,60]
Antibiotic resistance	2 [48,57]
Not reported	14 [29,32,33,35,36,37,40,43,45,47,59,62,63,68]

**Table 4 toxins-16-00373-t004:** Comparison of the mycotoxin concentrations found in genetically modified (GM) and non-GM maize. Data represent 13 studies containing 33 experiments. Concentrations are presented in mg of mycotoxin per Kg of maize.

		NT	T	
Analysis	Number of Articles ^2^	$\bar{X}$ ± SD (*n*)	$\bar{X}$ ± SD (*n*)	*p*
Total	13 [26,27,28,38,39,44,46,51,57,61,62,67,70]	4.16 ± 1.24 (954)	1.75 ± 0.45 (892)	<0.001
Total ^1^	13 [26,27,28,38,39,44,46,51,57,61,62,67,70]	2.06 ± 1.06 (906)	1.19 ± 0.38 (880)	<0.001
FUM	9 [26,38,39,44,46,51,61,67,70]	4.60 ± 0.64 (761)	1.88 ± 0.24 (726)	<0.001
FUM ^1^	9 [26,38,39,44,46,51,61,67,70]	2.52 ± 1.34 (713)	1.42 ± 0.46 (714)	<0.001
AFL	4 [26,27,28,67]	0.15 ± 0.03 (61)	0.08 ± 0.03 (34)	0.020
DON	2 [46,57]	0.70 ± 0.04 (90)	0.74 ± 0.06 (90)	<0.001
ZEA	1 [46]	0.01 ± <0.01 (42)	0.01 ± <0.01 (42)	<0.001

$\bar{X}$: mean (mg/Kg), SD: standard deviation (mg/Kg), *n* number of samples, *p*: *p*-value—inverse analysis of variance, NT: non-transgenic maize, T: transgenic maize, ^1^: there were discrepant results in the groups for the analyses, which were removed from the analysis in order to evaluate possible interference of these data in the results, ^2^: quantity of articles.

**Table 5 toxins-16-00373-t005:** Data on mycotoxins and the purpose of transgenics from the articles included in the meta-analysis.

Article ID	Transgenic Objective	Mycotoxin
Abbas, et al., 2005 [26]	Insect resistance	AFL, FUM
Abbas, et al., 2008 [27]	Insect resistance	AFL
Accinelli, et al., 2014 [28]	Insect resistance and herbicide resistance	AFL
Bordini, et al., 2019 [38]	Insect resistance and herbicide resistance	FUM
Bowers, et al., 2014 [39]	Insect resistance	FUM
Dowd, 2001 [44]	Insect resistance and herbicide resistance	FUM
Folcher, et al., 2010 [46]	Insect resistance	FUM, DON, ZEA
Herrera, et al., 2010 [51]	Insect resistance	FUM
Naef, et al., 2006 [57]	Insect resistance, herbicide resistance, and antibiotic resistance	DON
Rheeder, 2024 [61]	Insect resistance and herbicide resistance	FUM
Rocha, et al., 2016 [21]	Insect resistance	FUM
Weaver, et al., 2017 [66]	Insect resistance and herbicide resistance	AFL, FUM
YANG, et al., 2022 [69]	Insect resistance and herbicide resistance	FUM

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
