# Peer review of "Resistance of Transgenic Maize Cultivars to Mycotoxin Production—Systematic Review and Meta-Analysis"

_toxins, 2024, doi:10.3390/toxins16080373_

Round 1
Reviewer 1 Report
Comments and Suggestions for Authors
Dear Editors,
the work needs to be thoroughly revised.
The paper contains a lot of information that is very confusingly written and difficult for the reader to understand, especially the introduction and the results.
Point 2.1 The general findings, especially in lines 97 to 113, should be presented in a table to present the data better and make it easier to understand. In addition, the references are missing in this part. In addition, the authors cite "a large number of studies" in the paper (from line 115) without citing these studies. In the further course of the text, they also mention "three articles",... "three further articles",... which studies, why were they not cited?
Table 1 needs to be better described.
In Table 2 the papers are listed numerically, but again it is not known which papers they refer to as there are no references.
Table 3 again lists the number of works - but which works does this refer to??? They should be cited!!!
In the Materials and methods section, the PRISMA guidelines and meta-analysis should be briefly explained.
Figure 1, which refers to the diagram of the study, is at the end of the paper. The figure numbers need to be changed.
The conclusion also needs to be refined and the statement that the use of transgenic maize reduces the amount of mycotoxins by more than 50% needs to be clarified.
Kind regards!
Author Response
Comment 1: The paper contains a lot of information that is very confusingly written and difficult for the reader to understand, especially the introduction and the results.
Response 1: English and sentences were improved. The text was originally reviewed by a native-speaker (proof-reading service), and now it was further improved.
Comment 2: The general findings, especially in lines 97 to 113, should be presented in a table to present the data better and make it easier to understand. In addition, the references are missing in this part. In addition, the authors cite "a large number of studies" in the paper (from line 115) without citing these studies. In the further course of the text, they also mention "three articles",... "three further articles",... which studies, why were they not cited?
Response 2: A new table was added as suggested. Thanks for you suggestion.
Comment 3: Table 1 needs to be better described.
Response 3: Further description was added as suggested.
Comment 4: In Table 2 the papers are listed numerically, but again it is not known which papers they refer to as there are no references.
Response 4: All references were added. Thanks for pointing it out.
Comment 5: Table 3 again lists the number of works - but which works does this refer to??? They should be cited!!!
Response 5: References were added.
Comment 6: In the Materials and methods section, the PRISMA guidelines and meta-analysis should be briefly explained.
Response 6: Description for PRISMA in lines 435 to 444. Brief explanation was added.
Comment 7: Figure 1, which refers to the diagram of the study, is at the end of the paper. The figure numbers need to be changed.
Response 7: Figure numbers were reviewed and changed accordingly.
Comment 8: The conclusion also needs to be refined and the statement that the use of transgenic maize reduces the amount of mycotoxins by more than 50% needs to be clarified.
Response 8: The segment was reviewed and changed to better clarify our statements.
Reviewer 2 Report
Comments and Suggestions for Authors
I will have few comments to be addressed:
1. The first occurrence of a noun should be in full rather than abbreviation, such as AFL, FUM, DON, ZEA in the abstract, please check such problems in the full manuscript.
2. Some of the keywords are repeated, select other terms to increase discoverability.
3. The references are too old, and it is recommended to cite more references from the last five years.
4. The references are not properly cited, please cite the original references, for exemple, the first sentence of the paper does not cite the original references, please check such problems in the full manuscript.
5. In the discussion section, the author state that the production of mycotoxins are related to insects and temperature, and the arguments are insufficient, please supplement the relevant references and discuss it in detail.
Comments on the Quality of English LanguageModerate editing of English language required
Author Response
Comment 1: The first occurrence of a noun should be in full rather than abbreviation, such as AFL, FUM, DON, ZEA in the abstract, please check such problems in the full manuscript.
Response 1: All acronyms were reviewed. Thank you.
Comment 2: Some of the keywords are repeated, select other terms to increase discoverability.
Response 2: We added new Keywords as suggested. Thnaks.
Comment 3: The references are too old, and it is recommended to cite more references from the last five years.
Response 3: References were reviewed and updated. However, it was necessary to keep some older studies, since they are original articles and present the best information for our review.
Comment 4: The references are not properly cited, please cite the original references, for exemple, the first sentence of the paper does not cite the original references, please check such problems in the full manuscript.
Response 4: References were all reviewed and updated. Thanks for pointing it out.
Comment 5: In the discussion section, the author state that the production of mycotoxins are related to insects and temperature, and the arguments are insufficient, please supplement the relevant references and discuss it in detail.
Response 5: More detail was added to the text. Thank you for your suggestion.
Round 2
Reviewer 1 Report
Comments and Suggestions for Authors
Dear Authors,
the sentence referring to the results shown in Table 2 (line 111 to 121) should technically be moved in front of the table, not to the right of the table.
Kind regards!
Author Response
The sentence has been moved to the front of the table 2.
Reviewer 2 Report
Comments and Suggestions for Authors
All comments from my side had been responded and modified in the last version of the manuscript. My suggestion is that it could be accepted for publication.